# Modulation Awareness Method for Dual-Hop Cooperative Transmissions over Frequency-Selective Channels

**DOI:** 10.3390/s22145441

**Published:** 2022-07-21

**Authors:** Mohamed Marey, Hala Mostafa

**Affiliations:** 1Smart Systems Engineering Laboratory, College of Engineering, Prince Sultan University, Riyadh 11586, Saudi Arabia; 2Department of Information Technology, College of Computer and Information Sciences, Princess Nourah bint Abdulrahman University, P.O. Box 84428, Riyadh 11671, Saudi Arabia; hfmostafa@pnu.edu.sa

**Keywords:** modulation awareness, cooperative broadcasts, fading channels

## Abstract

Modulation awareness and cooperative transmissions have individually received a significant amount of research in the scholarly literature. However, a limited number of works are principally concerned with the combination of the two topics, and they are restricted to frequency-flat wireless channels. In this study, we propose a new modulation awareness method applicable to dual-hop amplify-and-forward cooperative broadcasts. The suggested method is built on the creation of theoretical representations of cross-correlation functions of the received signals. We conceptually prove that a family of modulation types generates spikes for certain cross-correlation functions, while others do not. We create a numerous layer hypothesis evaluation for the purpose of making judgments centered on this attribute. The suggested method has a number of benefits, such as the ability to operate on both frequency-flat and frequency-selective channels, as well as the absence of the necessity of channel awareness or noise power. Computer simulations analyze the performance of the proposed method, which delivers adequate awareness performance in a variety of operational scenarios.

## 1. Introduction

Automatic modulation awareness (AMA) was initially inspired by its implementation in military scenarios [1,2,3]. Electronic warfare and threat analysis are two examples of military uses that require the recognition of signal modulations in order to identify adversary transmitting units, to prepare jamming signals and to recover the intercepted signal. In addition, the adoption of modulation as an additional layer of protection makes it impossible for receivers to decipher a message unless they are aware of the specific modulation being used. The word “automatic” is employed in contrast to the first deployment of modulation classification, when signals were manually handled by skilled engineers using signal monitoring and processing instruments.

The majority of modulation classifiers developed over the previous three decades have been developed with the use of automated devices. This task has been given numerous terminologies in the literature, including automatic modulation classification, recognition, identification and awareness, all of which pertain to the same goal of knowing the modulation type at the receiver. It will be referred to as AMA throughout this work. Adaptive modulation is a method used by existing smart transceivers to improve the spectral efficiency and reliability by changing the modulation size while maintaining the same symbol rate (e.g., [4,5,6]). The basic principle is that the transmitter chooses, from a pool of possible modulations, the one that best accomplishes the required quality of service given the observed signal-to-noise ratio (SNR).

Herein lies the significance of AMA algorithms at receivers, since they have to know the kind of modulations used in order to decode the supplied data. Adaptive modulation is adopted in a number of current wireless technologies, such as cellular systems, microwave transmissions, satellites and wireless local area networks [7,8]. The likelihood-based (LB) and feature-based (FB) approaches are the most often used in the field of AMA [9,10]. In general, LB techniques tackle AMA issues using probability theories and hypothesis models. Even though LB AMA techniques reach the highest level of precision, they are computationally intensive and may be prohibitively costly in reality. To compensate for this limitation, FB AMA techniques exploit signal characteristics, such as higher order statistics and cyclostationarity to discriminate between modulation types [11,12,13,14].

Similarly, cooperative communications have sparked substantial attention in research and industry because of their superior performance characteristics such as system capacity and transmission range, including the simplicity with which they may be implemented [15,16,17,18]. The fundamental notion is driven by the fact that, in a wireless context, the signal sent by a source to a destination is also heard by other endpoints, which are frequently known as relays. The signals captured by the relays are recycled and sent to their destination. The destination subsequently integrates the messages from the source and the relays, resulting in spatial variety by leveraging numerous reception of the same data at different stations and transmission channels.

Extensive work has been made in information theoretic analysis of various cooperative systems to estimate feasible rates, capacity limitations and the diversity-multiplexing trade-off as well as data detection and channel estimation (e.g., [19,20,21,22,23,24]). Despite the importance of these findings, more work must be done before they can be put into practice. One of the primary concerns with cooperative systems is determining how to perform AMA in the context of multiple transmissions of the source and relays. Only two studies, to the best of our knowledge, have tackled the AMA issue for cooperative systems [10,25]. Nevertheless, they are restricted to frequency-flat fading channels and the decode-and-forward (DF) approach in their communication protocols.

The most important contributions and discoveries made by this research are described in the following aspects.

We develop a new AMA method for amplify-and-forward (AF) dual-hop cooperative systems over frequency-selective fading channels. Although adaptive modulation has been widely investigated in the literature (e.g., [26,27,28,29]), no work has been dedicated to performing modulation awareness for these systems over wireless selective channels. This confirms the novelty of this work. It is important to highlight the fact that AF cooperative systems are superior to DF cooperative systems in terms of capacity and processing.This study makes use of the temporal redundancy present in the structure of received signals to create correlation functions that include spikes for one set of modulations but not for others.With the assistance of this attribute, a multiple-layer AMA technique is constructed on the notion of false-alarms.

The proposed algorithm has the following advantages.

The suggested method does not require any information about the channel coefficients or the strength of the noise.The suggested method runs over any kind of wireless channels including frequency-flat and frequency-selective channels.The suggested method provides a simplified processing requirement.The suggested method does not required any pilot symbols to start the awareness process. This makes it appropriate for use in both military and civilian contexts.

The rest of this work is as follows. Section 2 describes the system model. Section 3 and Section 4 cover the cross-correlation functions for different modulation types as well as the method under consideration. Section 5 presents the results of the simulations. Section 6 contains the final conclusions.

## 2. System Model

We consider a three-terminal cooperative system as shown in Figure 1, where the source *S* has data for the delivery to the destination *D* with the aid of the relay *R*. To simplify the mathematical notations and analytical expressions, we assume that all nodes have a single antenna element. We leave the implementation of multiple antennas at each node to future development. The relay implements an AF protocol and runs in a half-duplex configuration that it transmits and receives at two distinct times [24].

At both the relay and the destination, we expect the additive white Gaussian noise (AWGN) variance to be $\sigma_{n}^{2}$. We denote $u=\left[u\left(0\right),u\left(1\right),\cdots ,u\left(K-1\right)\right]$ as a frame of transmission comprising *K* data symbols. Here, each data symbol is picked at random from a specific signal constellation α. The source selects α from a pool of candidates for each transmission frame to match the quality of service and capacity demands. A frame transmission will normally take place over two distinct stages.

During the first phase of the process, the frame u is sent via two distinct connections in order to reach both the relay and the final destination. We call $h_{SR}=\left[h_{SR}\left(0\right),\cdots ,h_{SR}\left(L-1\right)\right]$ and $h_{RD}=\left[h_{RD}\left(0\right),\cdots ,h_{RD}\left(L-1\right)\right]$ the channel impulse responses between *S* and *R* and between *R* and *D*, respectively. Here, *L* is the number of channel taps. We express the *k*th received samples at the relay, $z_{R}\left(k\right)$, and the destination, $z_{D}\left(k\right)$, analytically as
(1)$$z_{R}\left(k\right)=\sum_{l=0}^{L-1}h_{SR}\left(l\right)u\left(k-l\right)+n_{R}\left(k\right),$$
and
(2)$$z_{D}^{\left(1\right)}\left(k\right)=\sum_{l=0}^{L-1}h_{SD}\left(l\right)u\left(k-l\right)+n_{D}^{\left(1\right)}\left(k\right),$$
where $n_{R}\left(k\right)$ and $n_{D}^{\left(1\right)}\left(k\right)$ are the noise samples at the relay and destination, respectively. Here, we underline that both situations of direct connection presence and absence are extensively discussed in the literature (e.g., [19,20,21,22,23,24,25] and references therein). We focus on the existence of direct link in this investigation.

Phase two of transmission entails re-sending the processed the signal that was received by the AF relay. Accordingly, the input signal at the node *D* is written as
(3)$$z_{D}^{\left(2\right)}\left(k\right)=C\sum_{l=0}^{L-1}h_{RD}\left(l\right)z_{R}\left(k-l\right)+n_{D}^{\left(2\right)}\left(k\right),$$
where *C* is the scaling factor, and it is provided in the following form
(4)$$C=\frac{1}{\sqrt{\sum_{l=0}^{L-1}{\left|h_{SR}\left(l\right)\right|}^{2}+\sigma_{n}^{2}}}.$$

We set *C* to ensure that the output power of the AF relay and the source are identical. Plugging (Equation 1) into (Equation 3), we obtain
(5)$$z_{D}^{\left(2\right)}\left(k\right)=C\sum_{l=0}^{L-1}h_{RD}\left(l\right)\sum_{l^{\prime }=0}^{L-1}h_{SR}\left(l^{\prime }\right)u\left(k-l-l^{\prime }\right)+w_{D}^{\left(2\right)}\left(k\right),$$
where $w_{D}^{\left(2\right)}\left(k\right)$ is the *k*th noise sample and it incorporates both $n_{D}^{\left(2\right)}\left(k\right)$ and $n_{R}\left(k\right)$ contributions as
(6)$$w_{D}^{\left(2\right)}\left(k\right)=n_{D}^{\left(2\right)}\left(k\right)+C\sum_{l=0}^{L-1}h_{RD}\left(l\right)n_{R}\left(k-l\right).$$

The objective of this research is to devise a method for determining the allotted modulation type α by utilizing the features extracted from the signals $z_{D}^{\left(1\right)}\left(k\right)$ and $z_{D}^{\left(2\right)}\left(k\right)$, k=0,⋯,K−1.

## 3. Cross-Correlation Functions for Different Modulation Schemes

In this part, the cross-correlation functions for several modulation schemes are analyzed in an effort to pinpoint distinctive traits that may be utilized to identify these schemes. Across the whole of the exploration, we are going to operate on the following presumptions.

We employ the widely held premise in the literature that data symbols are unrelated to each other [30,31]. We write
(7)$$E\left[u\left(k_{1}\right)u\left(k_{2}\right)\right]=\left\{\begin{array}{cc} \delta \left(k_{1}-k_{2}\right) & \mathrm{for}\;\alpha =\mathrm{BPSK}\\ 0 & \mathrm{Otherwise}, \end{array}\right.$$
where $E\left[\cdot \right]$ is a random variable’s statistical expectation, $\delta \left(\cdot \right)$ is the function of Kronecker delta, and BPSK refers to a binary phase shift keying modulation. Furthermore, we compose
(8)$$E\left[u\left(k_{1}\right)u^{*}\left(k_{2}\right)\right]=\delta \left(k_{1}-k_{2}\right),$$
for all potential values of α pertain to the configuration choices of M−PSK and M−QAM. Here, *M* is the modulation order, and * denotes the complex conjugate operator. We suppose that the transmit power is one without sacrificing generality.The information symbols have no relationship with the noise values:
(9)$$\begin{array}{ccc} E\left[u\left(k_{1}\right)n_{R}\left(k_{2}\right)\right] & = & E\left[u\left(k_{1}\right)n_{R}^{*}\left(k_{2}\right)\right]=0,\\ E\left[u\left(k_{1}\right)n_{D}^{\left(1\right)}\left(k_{2}\right)\right] & = & E\left[u\left(k_{1}\right)n_{D}^{(1)*}\left(k_{2}\right)\right]=0,\\ E\left[u\left(k_{1}\right)n_{D}^{\left(2\right)}\left(k_{2}\right)\right] & = & E\left[u\left(k_{1}\right)n_{D}^{(2)*}\left(k_{2}\right)\right]=0. \end{array}$$The noise values are unrelated to one another:
(10)$$\begin{array}{ccc} E\left[n_{R}\left(k_{1}\right)n_{D}^{\left(1\right)}\left(k_{2}\right)\right] & = & E\left[n_{R}\left(k_{1}\right)n_{D}^{\left(2\right)}\left(k_{2}\right)\right]=0,\\ E\left[n_{R}\left(k_{1}\right)n_{D}^{(1)*}\left(k_{2}\right)\right] & = & E\left[n_{R}\left(k_{1}\right)n_{D}^{(2)*}\left(k_{2}\right)\right]=0,\\ E\left[n_{D}^{\left(1\right)}\left(k_{1}\right)n_{D}^{\left(2\right)}\left(k_{2}\right)\right] & = & E\left[n_{D}^{\left(1\right)}\left(k_{1}\right)n_{D}^{(2)*}\left(k_{2}\right)\right]=0. \end{array}$$The parameters of the channel are initialized at random, and they remain constant throughout the frame.

We describe the following correlation functions:
(11a)$$F_{1}\left(k_{1},k_{2}\right)=E\left[z_{D}^{\left(1\right)}\left(k_{1}\right)z_{D}^{\left(2\right)}\left(k_{2}\right)\right],$$
(11b)$$F_{2}\left(k_{1},k_{2}\right)=E\left[z_{D}^{\left(1\right)}\left(k_{1}\right){\left(z_{D}^{\left(2\right)}\left(k_{2}\right)\right)}^{3}\right],$$
(11c)$$F_{3}\left(k_{1},k_{2}\right)=E\left[z_{D}^{\left(1\right)}\left(k_{1}\right){\left(z_{D}^{\left(2\right)}\left(k_{2}\right)\right)}^{7}\right],$$
and
(11d)$$F_{4}\left(k_{1},k_{2}\right)=E\left[{\left(z_{D}^{\left(1\right)}\left(k_{1}\right)\right)}^{4}{\left(z_{D}^{(2)*}\left(k_{2}\right)\right)}^{2}\right].$$

We develop the following expressions with the assistance of Table 1 and the aformentioned assumptions.
(12a)$$F_{1}\left(k_{1},k_{2}\right)=\left\{\begin{array}{cc} C\sum_{l_{1},l_{2},l_{3}=0}^{L-1}h_{SD}\left(l_{1}\right)h_{RD}\left(l_{2}\right)h_{SR}\left(l_{3}\right)\delta \left(k_{1}-k_{2}-l_{1}+l_{2}+l_{3}\right) & \mathrm{for}\;\alpha =\mathrm{BPSK}\\ 0 & \mathrm{otherwise}, \end{array}\right.$$
(12b)$$F_{2}\left(k_{1},k_{2}\right)=\left\{\begin{array}{cc} C^{3}\;\;\;\;\;\sum_{l_{1},l_{2},\cdots ,l_{7}=0}^{L-1}\;\;\;\;\;h_{SD}\left(l_{1}\right)\prod_{i=2}^{4}h_{RD}\left(l_{i}\right)\prod_{i^{\prime }=5}^{7}h_{SR}\left(l_{i^{\prime }}\right)\delta \left(k_{1}\;-\;k_{2}\;-l_{1}\;+l_{2}\;+\;\cdots \;+\;l_{7}\right) & \mathrm{for}\;\alpha =\mathrm{BPSK},\;\mathrm{QPSK},\\ -0.68C^{3}\;\;\;\;\;\sum_{l_{1},l_{2},\cdots ,l_{7}=0}^{L-1}\;\;\;\;\;h_{SD}\left(l_{1}\right)\prod_{i=2}^{4}h_{RD}\left(l_{i}\right)\prod_{i^{\prime }=5}^{7}h_{SR}\left(l_{i^{\prime }}\right)\delta \left(k_{1}\;-\;k_{2}\;-l_{1}\;+l_{2}\;+\;\cdots \;+\;l_{7}\right) & \mathrm{for}\;\alpha =16-\mathrm{QAM},\\ -0.619C^{3}\;\;\;\;\;\sum_{l_{1},l_{2},\cdots ,l_{7}=0}^{L-1}\;\;\;\;\;h_{SD}\left(l_{1}\right)\prod_{i=2}^{4}h_{RD}\left(l_{i}\right)\prod_{i^{\prime }=5}^{7}h_{SR}\left(l_{i^{\prime }}\right)\delta \left(k_{1}\;-\;k_{2}\;-l_{1}\;+l_{2}\;+\;\cdots \;+\;l_{7}\right) & \mathrm{for}\;\alpha =64-\mathrm{QAM},\\ 0 & \mathrm{for}\;\alpha =8-\mathrm{PSK},\;16-\mathrm{PSK}, \end{array}\right.$$
(12c)$$F_{3}\left(k_{1},k_{2}\right)=\left\{\begin{array}{cc} C^{7}\;\;\;\;\;\sum_{l_{1},l_{2},\cdots ,l_{15}=0}^{L-1}\;\;\;\;\;h_{SD}\left(l_{1}\right)\prod_{i=2}^{8}h_{RD}\left(l_{i}\right)\prod_{i^{\prime }=9}^{15}h_{SR}\left(l_{i^{\prime }}\right)\delta \left(k_{1}\;-\;k_{2}\;-l_{1}\;+l_{2}\;+\;\cdots \;+\;l_{15}\right) & \mathrm{for}\;\alpha =\mathrm{BPSK},\;\mathrm{QPSK},\;8-\mathrm{PSK}\\ 2.2C^{7}\;\;\;\;\;\sum_{l_{1},l_{2},\cdots ,l_{15}=0}^{L-1}\;\;\;\;\;h_{SD}\left(l_{1}\right)\prod_{i=2}^{8}h_{RD}\left(l_{i}\right)\prod_{i^{\prime }=9}^{15}h_{SR}\left(l_{i^{\prime }}\right)\delta \left(k_{1}\;-\;k_{2}\;-l_{1}\;+l_{2}\;+\;\cdots \;+\;l_{15}\right) & \mathrm{for}\;\alpha =16-\mathrm{QAM},\\ 1.91C^{7}\;\;\;\;\;\sum_{l_{1},l_{2},\cdots ,l_{15}=0}^{L-1}\;\;\;\;\;h_{SD}\left(l_{1}\right)\prod_{i=2}^{8}h_{RD}\left(l_{i}\right)\prod_{i^{\prime }=9}^{15}h_{SR}\left(l_{i^{\prime }}\right)\delta \left(k_{1}\;-\;k_{2}\;-l_{1}\;+l_{2}\;+\;\cdots \;+\;l_{15}\right) & \mathrm{for}\;\alpha =64-\mathrm{QAM},\\ 0 & \mathrm{for}\;\alpha =16-\mathrm{PSK}, \end{array}\right.$$
and
(12d)$$F_{4}\left(k_{1},k_{2}\right)=\left\{\begin{array}{cc} C^{2}\sum_{l_{1},l_{2},\cdots ,l_{8}=0}^{L-1}\prod_{i=1}^{2}h_{SR}^{*}\left(l_{1}\right)\prod_{i^{\prime }=3}^{4}h_{RD}^{*}\left(l_{i^{\prime }}\right)\prod_{i^{\prime \prime }=5}^{8}h_{SD}\left(l_{i^{\prime \prime }}\right)\delta \left(k_{1}\;-\;k_{2}\;-l_{1}\;+l_{2}\;+\;\cdots \;+\;l_{8}\right) & \mathrm{for}\;\alpha =\mathrm{BPSK},\;\mathrm{QPSK},\\ 0\;\;\;\;\;\;\;\;\;\;\;\;\;\mathrm{for}\alpha =8-\mathrm{PSK},\;16-\mathrm{PSK}, & 16-\mathrm{QAM},\;64-\mathrm{QAM} \end{array}\right.$$

Figure 2, Figure 3, Figure 4 and Figure 5 illustrate snapshots of the magnitudes of the previous functions for different modulation schemes at SNR = 15 dB. We observed that each correlation function displays spikes for one modulation scheme but no spikes for the other modulation scheme. The spikes are placed around the diagonal, $|k_{1}-k_{2}|\geq L$, and their amplitudes are controlled by the channel coefficients. This is consistent with the mathematical findings presented in the prior discussions. For the sake of presentation, we selected a multipath channel with five taps.

However, the mathematical developments revealed that the suggested method is generic in the sense that it is applicable to any wireless channel that is of any length. In addition, the proposed method does not require any prior knowledge of the channel’s length or statistics as explained in the following section. The remaining simulation settings are stated in Section 5. Figure 6 provides a high-level overview of the structure of a cross-correlation function when spikes are present. In reality, the restricted monitoring interval allows nonzero spikes to appear where there should be zeros. However, after a lengthy time of observation, they are statistically insignificant.

## 4. Proposed Method

The suggested method takes use of the existence of spikes in the previously described cross-correlation functions for modulation awareness. For the sake of demonstration, we take a look at the set of modulation schemes {QPSK, 16 − PSK, 64 − QAM}. When $F_{2}\left(k_{1},k_{2}\right)$ has spiked, the set of {QPSK, 64 − QAM} is said to possess; otherwise, the 16 − PSK modulation scheme is selected. Finally, we examine the spikes of $F_{4}\left(k_{1},k_{2}\right)$ to distinguish between 64 − QAM and QPSK; if they exist, QPSK is reported present; otherwise, 64 − QAM is accepted.

For spike detection, the accompanying fundamental considerations must be addressed.

In reality, the observation duration is constrained, which prompts the question of how to calculate the cross-correction functions.Given the preceding fact, an estimating error of $F_{i}\left(k_{1},k_{2}\right)$, i=1,⋯,4, outcomes a non-zero level when $F_{i}\left(k_{1},k_{2}\right)$ should remain zero. This has a negative impact on the conclusion of whether or not statistical peaks of $F_{i}\left(k_{1},k_{2}\right)$ exist. Due to this, judgment criteria have to be established in situations where a non-ideal predictor is present, even if previous information of the estimate error statistics is not available.It is also assumed that channel information and noise power are unknown. This assumption constitutes an extra obstacle in the way of successfully completing the spike detection task.

To determine the most probable assigned modulation scheme, we offer a spike-detection method bound by the chance of false alarm while dealing with the aforementioned constraints. The fundamental concept is to calculate a test statistic and establish a barrier. If the test statistic exceeds the barrier, the method asserts that spikes exist; otherwise, no spikes are discovered. The explanation that follows gives a more explicit mathematical analysis of the suggested method.

Let us examine a number of successive samples, *K*, from each of the two vectors $z_{D}^{\left(1\right)}\left(k\right)$ and $z_{D}^{\left(2\right)}\left(k\right)$, k=0,⋯,K−1. When spikes appear in a correlation function, they tend to cluster around the diagonal, as demonstrated in Figure 2, Figure 3, Figure 4, Figure 5 and Figure 6. The test statistic μ is defined as
(13)$$\mu =\frac{1}{K}\left|\sum_{k=0}^{K-1}f\left(z_{D}^{\left(1\right)}\left(k\right),z_{D}^{\left(2\right)}\left(k\right)\right)\right|,$$
where
(14)$$f\left(z_{D}^{\left(1\right)}\left(k\right),z_{D}^{\left(2\right)}\left(k\right)\right)=\left\{\begin{array}{cc} z_{D}^{\left(1\right)}\left(k\right)z_{D}^{\left(2\right)}\left(k\right) & \mathrm{if}F_{1}\mathrm{is}\mathrm{considered}\\ z_{D}^{\left(1\right)}\left(k\right){\left(z_{D}^{\left(2\right)}\left(k\right)\right)}^{3} & \mathrm{if}F_{2}\mathrm{is}\mathrm{considered}\\ z_{D}^{\left(1\right)}\left(k\right){\left(z_{D}^{\left(2\right)}\left(k\right)\right)}^{7} & \mathrm{if}F_{3}\mathrm{is}\mathrm{considered}\\ {\left(z_{D}^{\left(1\right)}\left(k\right)\right)}^{4}{\left(z_{D}^{(2)*}\left(k\right)\right)}^{2} & \mathrm{if}F_{4}\mathrm{is}\mathrm{considered}. \end{array}\right.$$

There are two hypotheses to investigate in the spike detection challenge:(15)$$\begin{array}{ccc} H_{0}:\;\mu & = & e,\\ H_{1}:\;\mu & = & S+e, \end{array}$$
where $H_{1}$ and $H_{0}$ are the hypotheses related to whether or not a spike exists. Here, *S* represents the spike’s amplitude and *e* denotes the impact of the AWGN and estimate error due to the usage of a restricted number of samples.

Using hypothesis $H_{0}$ and the assumption that *e* is a Rayleigh random variable, with mean $(\pi \sigma^{2})/2$ and cumulative distribution function (CDF) supplied by
(16)$$g\left(a\right)=1-\exp\left(-a^{2}/2\sigma^{2}\right),\;a\geq 0.$$

The method’s primary idea is to create a barrier λ, and the spike is stated to appear if μ>λ; alternatively, there would be no spike. Our aim is to achieve a given level of false alarm probability $P_{f}$, which is described as the possibility of incorrectly reporting the presence of a spike. As a consequence, one may write
(17)$$P_{f}=\mathrm{Pr}\left(\mu \geq \lambda |H_{0}\right)=1—g\left(\lambda \right).$$

By using (Equation 16) into (Equation 17), we obtain
(18)$$\lambda ={\left(-2\sigma^{2}\ln P_{f}\right)}^{0.5}.$$

It is transparent from (Equation 18) that, in order to obtain λ, the amount of $\sigma^{2}$ must be known. We compute $\sigma^{2}$ by calculating observations of $z_{D}^{\left(1\right)}\left(n\right)$ and $z_{D}^{\left(2\right)}\left(n\right)$ as
(19)$$\begin{array}{ccc} B & = & \left|\frac{1}{K(K-2L)}\sum_{\begin{array}{c} k_{1},k_{2}=0,\\ \left|k_{1}-k_{2}\right|\geq L \end{array}}^{K-1}f\left(z_{D}^{\left(1\right)}\left(k_{1}\right),z_{D}^{\left(2\right)}\left(k_{2}\right)\right)\right|, \end{array}$$
where $f\left(z_{D}^{\left(1\right)}\left(n\right),z_{D}^{\left(2\right)}\left(n\right)\right)$ is defined in (Equation 14). Considering that *B* is a Rayleigh random variable with a mean of $\sigma \sqrt{\pi }/2$, the computed level $\sigma^{2}$ can be
(20)$${\hat{\sigma }}^{2}=\frac{2B^{2}}{\pi }.$$

The computational complexity of the suggested AMA method is analysed by counting the number of necessary floating point operations (fops) [32,33]. The multiplication of two complex numbers takes six fops, while their addition only requires two fops. A complex number’s modulus involves three fops. Based on the proceeding analysis, the number of fops required at each node is approximately expressed as $18(K^{2}+K)$. Available technology appears to be sufficient for the algorithm’s practical execution. For a numerical example, we assume that *K* = 1000 with a processor of 10 Terafops per second, the processing time is 18 microsecond, which is feasible for real-world applications.

## 5. Simulation Results

Computer simulations have been conducted in order to evaluate how well the proposed method works. The settings of the system are as follows, unless otherwise specified: the number of symbols that are transmitted in each frame is equal to K = 10,000, and the likelihood of a false alert is set at $P_{f}=0.01$. In each of the wireless links, the discrete time channel impulse response was constructed using L=5 channel coefficients. The channel impulse responses have power delay profiles with exponential decay as [30,34]
(21)$$\sigma_{hj}^{2}\left(l\right)=X_{hj}\exp\left(-l/9\right),$$
where l=0,⋯,4, $h\in \left\{S,R\right\}$, $j\in \left\{R,D\right\}$. $X_{hj}$ is computed in order to account for the effect of path-loss on the awareness performance as
(22)$$X_{hj}={\left(\frac{w_{SD}}{w_{hj}}\right)}^{\rho },$$
where ρ refers to the path-loss exponent, $w_{hj}$ is the distance between terminals *h* and *j*, and nodes *S* and *D* are separated by a distance denoted by $w_{SD}$. For the purposes of the simulations, we set ρ=3.4, $w_{SD}=1$, $w_{SR}=0.43$ and $w_{RD}=0.56$. It is critical to remember that all distances have been calibrated to $w_{SD}$. The probability of proper awareness, $P_{cc}\left(\alpha =\xi |\xi \right)$, is employed as a quality criterion for the proposed method, where ξ is one of modulation options.

There are four different sets of modulation forms that are looked at as follows. $\theta_{1}=\left\{\mathrm{BPSK},\;\mathrm{QPSK}\right\}$, $\theta_{2}=\left\{\mathrm{BPSK},\;\mathrm{QPSK},8-\mathrm{PSK}\right\}$, $\theta_{3}=\left\{\mathrm{QPSK},16-\mathrm{PSK},16-\mathrm{QAM}\right\}$, and $\theta_{4}=\left\{\mathrm{BPSK},\;\mathrm{QAM},8-\mathrm{PSK},64-\mathrm{QAM}\right\}$. It is important to note that the proposed method is general in the sense that it can be applied to any number of modulation schemes as long as there are correlation functions that show spikes for some modulation schemes but not for others. There are a total of one thousand Monte Carlo attempts. It is worth noting that we only use the cross-correlation functions that are needed for the modulation of interest.

The function $F_{1}$ is employed for the set $\theta_{1}$; the functions $F_{1}$ and $F_{2}$ are used for the set $\theta_{2}$; the functions $F_{2}$ and $F_{3}$ are used for the set $\theta_{3}$; and the functions $F_{1}$, $F_{2}$ and $F_{3}$ are utilized for the set $\theta_{4}$. For example, the AMA for $\theta_{4}$ was performed in the following manner. The proposed statistical test on the function $F_{1}$ distinguishes BPSK modulation from other modulation techniques. The presence of BPSK is marked by the appearance of spikes, while the lack of spikes indicates the presence of other modulation schemes.

Then, using the proposed statistical test on the function $F_{2}$, we distinguish the 8 − PSK modulation scheme from the QPSK and 64 − QAM schemes. The absence of spikes indicates 8 − PSK modulation systems, but the presence of spikes indicates QPSK or 64 − QAM modulation schemes. Finally, we apply the proposed statistical test on the function $F_{3}$ to classify between the QPSK and 64 − QAM modulation schemes. In this case, the loss of spikes implies the 64 − QAM modulation scheme, but the presence of spikes shows the QPSK modulation scheme.

Figure 7, Figure 8, Figure 9 and Figure 10 depict the awareness performance for the modulation groups $\theta_{1}$, $\theta_{2}$, $\theta_{3}$ and $\theta_{4}$, respectively, as a function of SNR. The results indicate that the awareness performance of QPSK, 8 − PSK, 16 − PSK and 8 − PSK shown in Figure 7, Figure 8, Figure 9 and Figure 10, respectively, are not impacted in any way by the SNR values. This is as a result of the fact that their performance is determined by the probability of a false alarm, which was planned to be 0.01. On the other hand, the other modulation choices exhibit significant improvements in performance as the SNR value rises. This is due to the fact that as the SNR levels are increased, the spike detection performance is strengthened.

Figure 11 and Figure 12 show the impact of $P_{f}$ on the awareness performance for the modulation set $\theta_{1}$. As SNR improves, the findings show that the awareness performance dependency on correctly detecting the BPSK modulation scheme on $P_{f}$ diminishes. The justification for this is that, when SNR is larger, the influence of the noise contribution becomes less noticeable, and the performance is determined mostly by the estimate error that arises from using a limited observation time. As the observation time approaches infinity, this error disappears. In principle, reducing $P_{f}$ leads to a decrease in the barrier value, which increases the BPSK modulation scheme’s efficiency as indicated in Figure 13. The distinction of the QPSK modulation scheme, on the other hand, is the inverse. As a result, $P_{f}$ is intended to accomplish a performance compromise for BPSK and QPSK awareness.

Table 2 illustrates the confusion matrices for the modulation set $\theta_{4}$ at different values of the number of symbols that are transmitted in a frame, *N*. When *N* is increased from 2000 to 10,000, there is a discernible rise in the level of performance associated with awareness. On the other hand, there is no significant improvement in performance beyond *N* = 10,000. This represents the fact that the impact of the estimating error becomes less significant as the number of observations increases.

For comparative purposes, Figure 14 depicts the awareness performance obtained by the proposed method in addition to that obtained by the technique described in [10,25] for the modulation set $\theta_{4}$. The findings reveal that the proposed method significantly outperforms the methods described in [10,25], which are unable to deliver a suitable awareness performance even at high SNR levels. This is because the approaches described in [10,25] are based on a DF relaying scheme in which the relay re-transmits the received frame if it is correctly decoded. Here, we recall that the DF relaying system may fail to decode the received frame at the relay due to channel circumstances, which has a detrimental impact on modulation awareness performance. Furthermore, the method presented in [25] demonstrates a performance with poor awareness due to the absence of a training preamble.

## 6. Conclusions

The topic of modulation awareness for amplify-and-forward dual-hop cooperative systems over frequency-selective channels was explored in this work. A variety of cross-correlation functions between the received signals were given to create the groundwork for the suggested method. Mathematical proofs and simulation results for specific modulations revealed fixed location spikes in a number of cross-correlation functions; however, no such spikes were detected for the other kinds of modulation. This discovery was used to guide the awareness process through a tree-based method utilizing the false-alarm criterion for spike detection. The outcomes showed that good awareness performance was revealed in a range of functional circumstances. These fantastic results were produced with no prior knowledge of the channel status information or noise power.

## Figures and Tables

**Figure 1 sensors-22-05441-f001:**
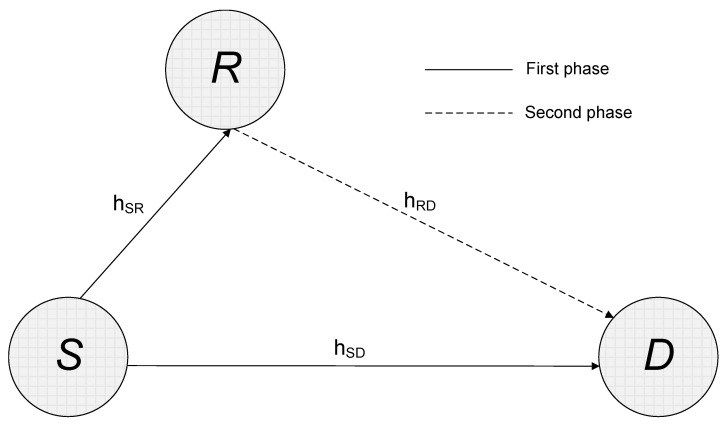
The three-terminal cooperative system under consideration.

**Figure 2 sensors-22-05441-f002:**
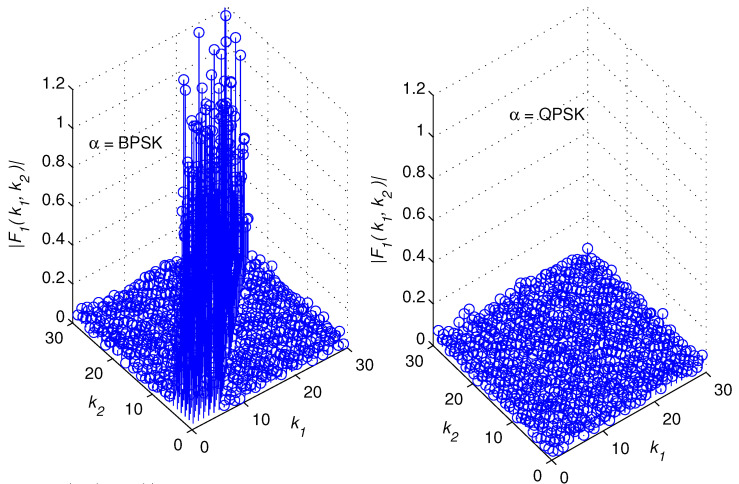
$\left|F_{1}\left(k_{1},k_{2}\right)\right|$ for BPSK and QPSK modulations.

**Figure 3 sensors-22-05441-f003:**
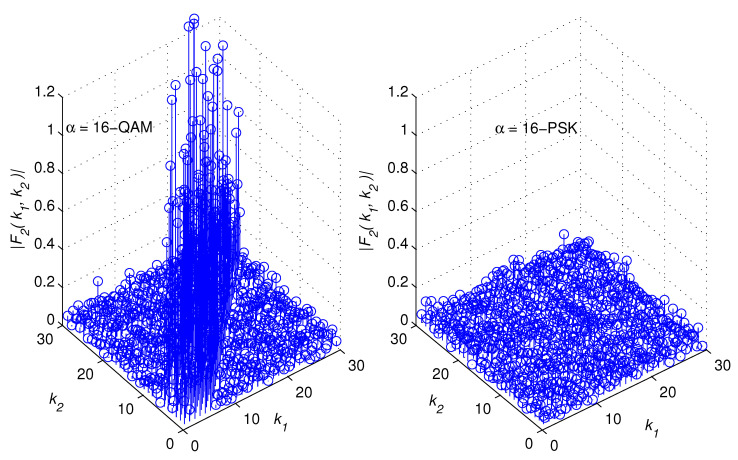
$\left|F_{2}\left(k_{1},k_{2}\right)\right|$ for 16 − QAM and 16 − PSK modulations.

**Figure 4 sensors-22-05441-f004:**
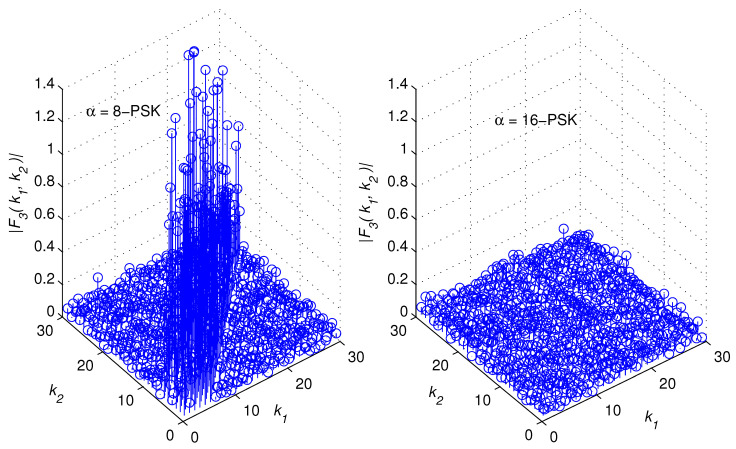
$\left|F_{3}\left(k_{1},k_{2}\right)\right|$ for 8 − PSK and 16 − PSK modulations.

**Figure 5 sensors-22-05441-f005:**
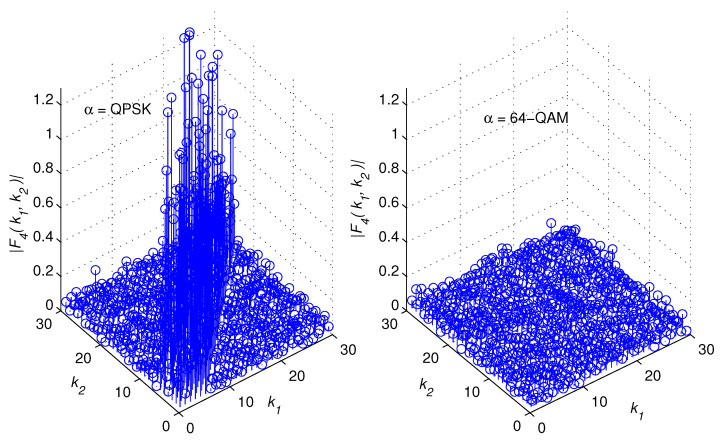
$\left|F_{4}\left(k_{1},k_{2}\right)\right|$ for QPSK and 64 − QAM modulations.

**Figure 6 sensors-22-05441-f006:**
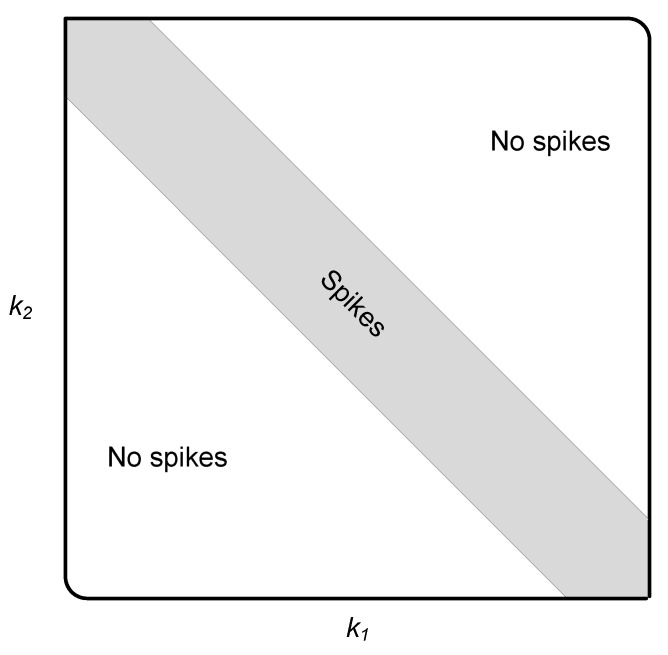
The architecture of a cross-correlation function when there are spikes.

**Figure 7 sensors-22-05441-f007:**
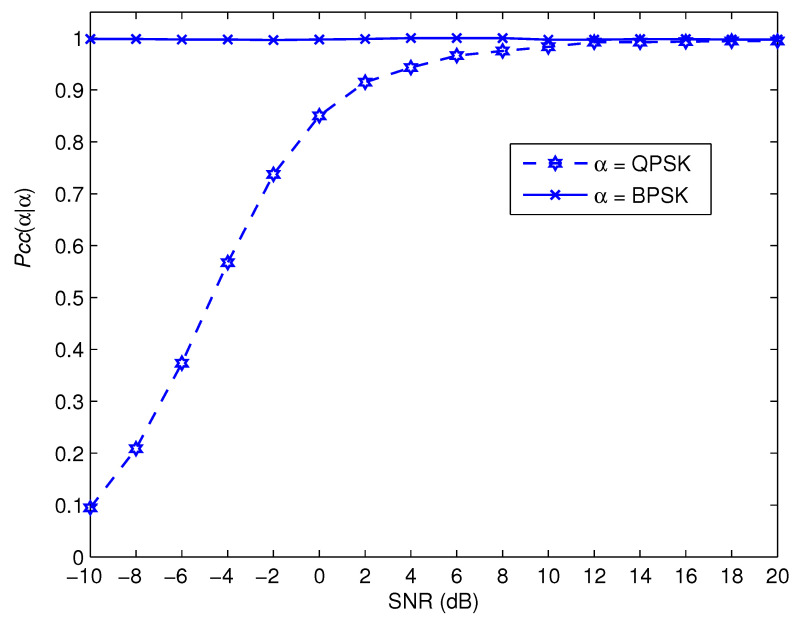
$Pcc\left(\alpha |\alpha \right)$ for the modulation group of $\theta_{1}$.

**Figure 8 sensors-22-05441-f008:**
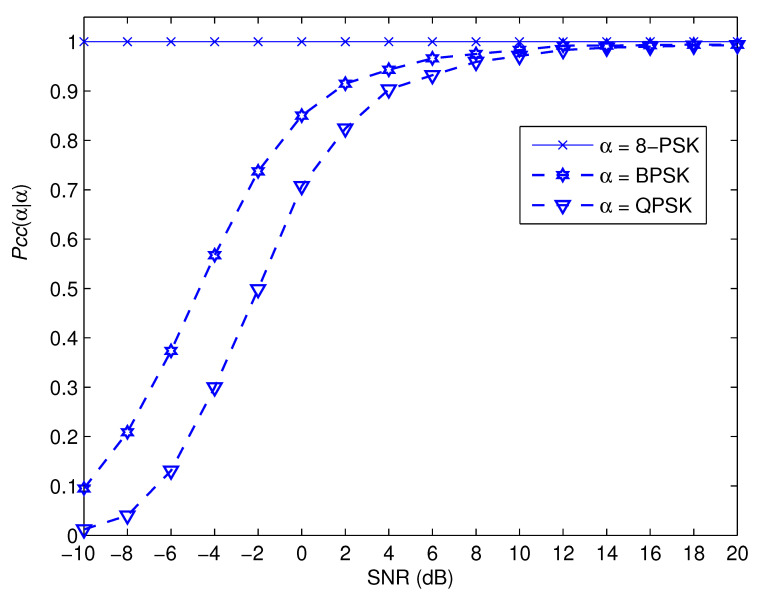
$Pcc\left(\alpha |\alpha \right)$ for the modulation group of $\theta_{2}$.

**Figure 9 sensors-22-05441-f009:**
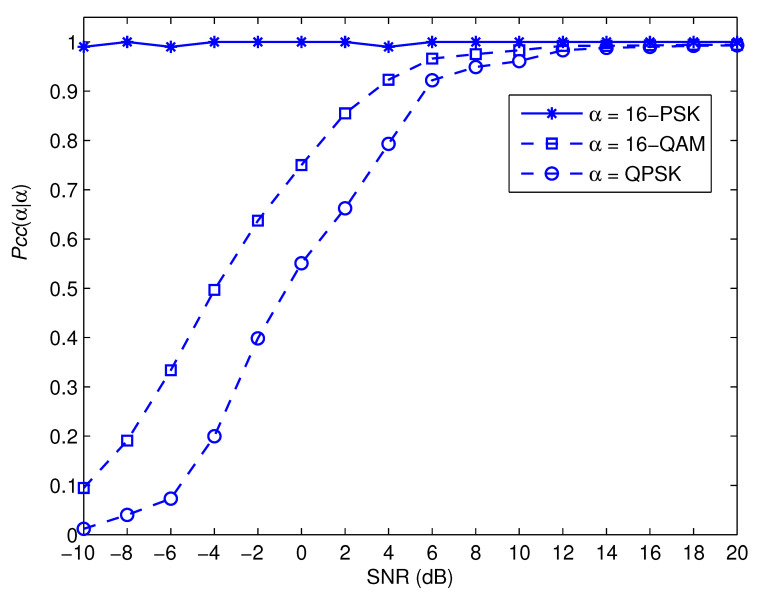
$Pcc\left(\alpha |\alpha \right)$ for the modulation group of $\theta_{3}$.

**Figure 10 sensors-22-05441-f010:**
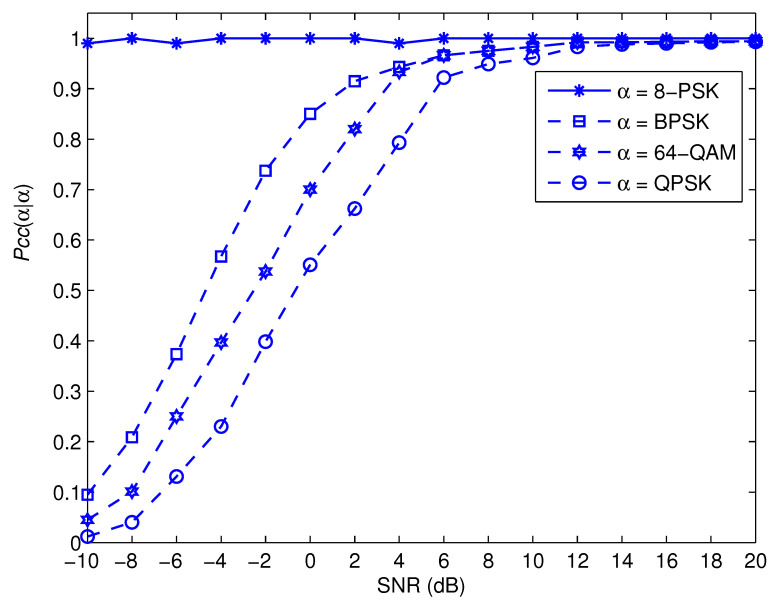
$Pcc\left(\alpha |\alpha \right)$ for the modulation group of $\theta_{4}$.

**Figure 11 sensors-22-05441-f011:**
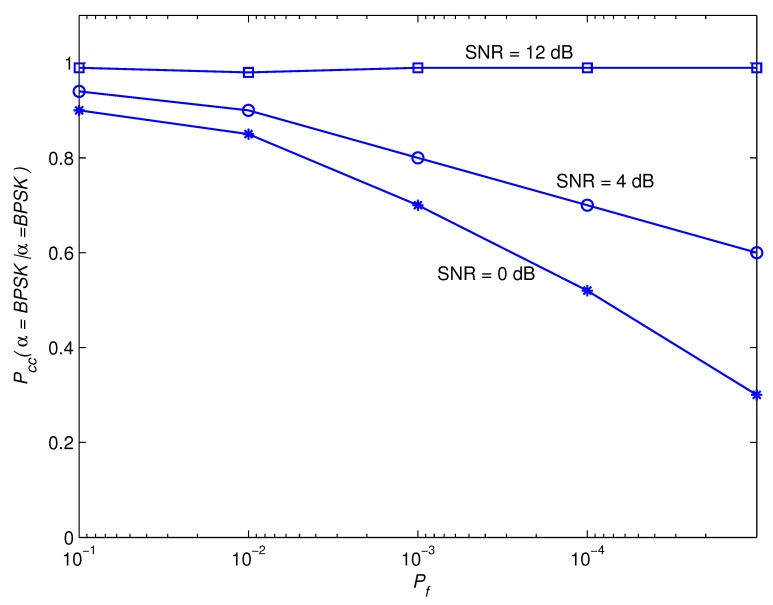
$Pcc\left(\alpha =\mathrm{BPSK}|\alpha =\mathrm{BPSK}\right)$ for the modulation group of $\theta_{1}$.

**Figure 12 sensors-22-05441-f012:**
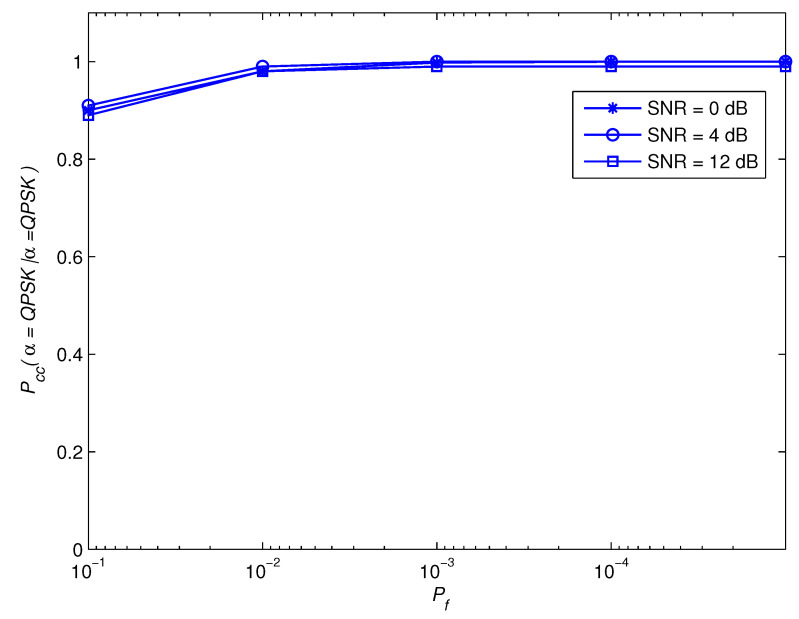
$Pcc\left(\alpha =\mathrm{QPSK}|\alpha =\mathrm{QPSK}\right)$ for the modulation group of $\theta_{1}$.

**Figure 13 sensors-22-05441-f013:**
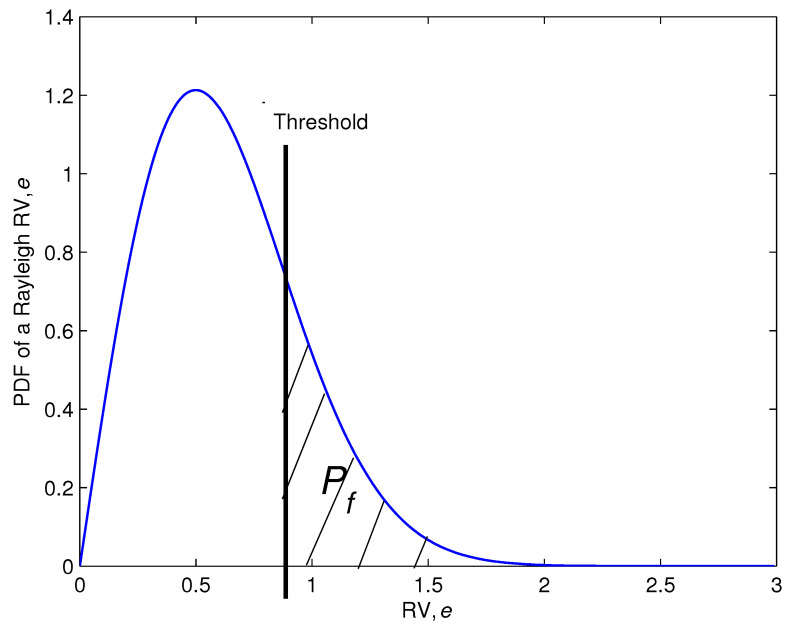
PDF of the Rayleigh RV, *e*.

**Figure 14 sensors-22-05441-f014:**
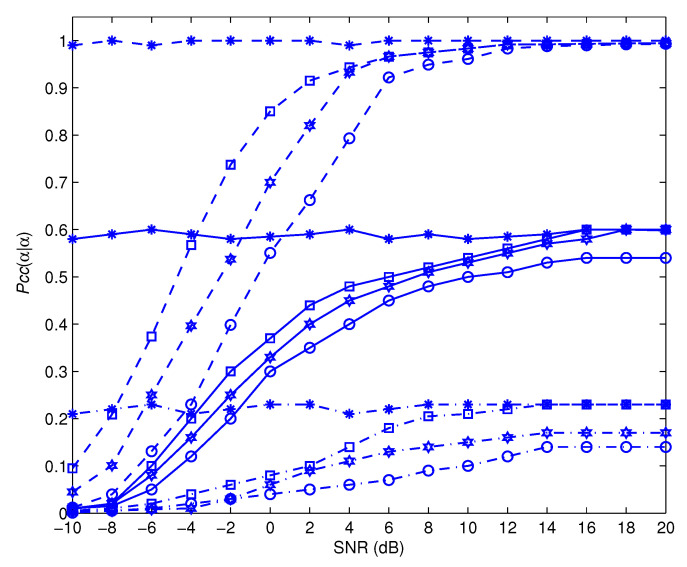
Awareness performance comparison with the modulation set $\theta_{4}$. Dashed lines represent the proposed method, sold lines are used for the method of [10], and dot-dashed are used for the method of [25]. Star, square, hexagram, and circle markers refer to 8-PSK, BPSK, 64 − QAM, and QPSK modulations, respectively.

**Table 1 sensors-22-05441-t001:** Higher order expectations for different modulation schemes with unit power.

	BPSK	QPSK	8-PSK	16-PSK	16-QAM	64 − QAM
$E\left[u^{2}\left(k\right)\right]$	1	0	0	0	0	0
$E\left[u^{4}\left(k\right)\right]$	1	1	0	0	−0.68	−0.619
$E\left[u^{8}\left(k\right)\right]$	1	1	1	0	2.2	1.91
$E\left[u^{6}\left(k\right)u^{*2}\left(k\right)\right]$	1	1	0	0	0	0

**Table 2 sensors-22-05441-t002:** Confusion matrices at different number of received symbols, *K*, at SNR = 15 dB.

*K* = 2000					*K* = 6000				
	BPSK	QPSK	8-PSK	64 − QAM		BPSK	QPSK	8-PSK	64 − QAM
BPSK	0.652	0.213	0.056	0.079	BPSK	0.852	0.113	0.026	0.079
QPSK	0.21	0.585	0.156	0.049	QPSK	0.184	0.721	0.056	0.039
8-PSK	0.058	0.28	0.551	0.111	8-PSK	0.038	0.225	0.721	0.016
64 − QAM	0.017	0.013	0.04	0.93	64 − QAM	0.01	0.01	0.02	0.96
*K* = 10,000					*K* = 12,000				
	BPSK	QPSK	8-PSK	64 − QAM		BPSK	QPSK	8-PSK	64 − QAM
BPSK	0.99	0.007	0.002	0.001	BPSK	1	0	0	0
QPSK	0.009	0.980	0.003	0.008	QPSK	0.001	0.999	0	0
8-PSK	0.007	0.002	0.983	0.008	8-PSK	0	0.002	0.996	0.002
64 − QAM	0.009	0.002	0.002	0.987	64 − QAM	0.002	0.003	0.004	0.991

## Data Availability

Not applicable.

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
