# Peer review of "Modulation Awareness Method for Dual-Hop Cooperative Transmissions over Frequency-Selective Channels"

_sensors, 2022, doi:10.3390/s22145441_

Round 1

Reviewer 1 Report

While I am not an expert in signal classification and do not have a good basis to judge the originality of this work, I am aware of rather sophisticated methods involving cyclostationary signal processing (CSP) that involve correlation of higher moments (cyclic cumulants).  There is no reference to CSP classification techniques in the introduction, so I point this out as something the authors may wish to consider including.  That aside, I found this paper well written, interesting, and with no obvious errors, and therefore I recommend that it be published. My main concern is that the testing of the method is restricted to single representation of the channel impulse response function (IRF), which is a single decaying exponential with an e-folding width of 9 samples, but is only sampled out to 5 samples.  I believe this is the reason that the band of spikes in the cross-correlation functions cuts off abruptly at a half-width of 5 samples, which certainly aids the success of the threshold detection.  I would like to see this additional dimension explored; how does the classification method perform over a range of IRF widths and also for multipath environments where the IRF is a sum of delayed versions of the single-mode IRF with different amplitudes.  This does not have to be done within the context of this paper, but could be reported in a future publication.  It seems that these generalities could be fairly easily added to the computational framework assembled for this Monte Carlo study.

Author Response

Please see the attached response. 

Author Response

Please see the attached response. 

Reviewer 3 Report

In this paper, the authors propose a new modulation awareness method applicable to dual-hop amplify-and-forward cooperative broadcasts. The suggested method is built on the creation of theoretical representations of cross-correlation functions of the received signals. Computer simulations are utilized in order to analyze and investigate the performance of the aforementioned method. The findings provide evidence that the proposed method delivers adequate results in a variety of operational scenarios. However, I still have some concerns as follows.

1. Literature review is not sufficient. The related work part is missing. A short survey about modulation awareness and adaptive communication should be conducted.

2. The contributions in the paper should be enhanced and presented clearly. Compared with existing works, what are the advantages of the methods proposed in this paper?

3. The benefits and applications of the proposed method should be enhanced. More analysis and discussion should be provided.

4. There are more opportunities for conducting meaningful experiments to comprehensively evaluate the method performance, such as more comparison experiments with many other existing methods.

5. More references about adaptive communication and cross-correlation analysis should be investigated. For example:

- Adaptive communication protocols in flying ad hoc network, IEEE Communications Magazine.

- Smartphone-assisted smooth live video broadcast on wearable cameras, Proc. of IEEE/ACM International Symposium on Quality of Service.

- A distributed swarm optimizer with adaptive communication for large-scale optimization. IEEE transactions on cybernetics.

- SCoP: Smartphone energy saving by merging push services in fog computing, Proc. of IEEE/ACM International Symposium on Quality of Service.

Author Response

Please see the attached response. 

Reviewer 4 Report

This work studies MA aspect in AF-based systems with frequency-selective channels. The work is generally interesting. However, the limited contribution and poor presentation prevent me from accepting the paper in the current form. My detailed comments are listed below:

1. The introduction can be improved. For instance, the motivation for using MA in an AF relay system (instead of DF) and consideration of the frequency-selective environment is missing. In other words, the importance/contribution of the work is not clear.

2. The source, relay, and destination all have a single antenna, which significantly limits the contribution of the work.

3. In the system model, is the direct link (h_sd) being considered? If so (or not), please motivate.

4. Please motivate the key assumption of the paper on Page 4: "the various data symbols do not correlate with one another in any way".

5. Please elaborate more on the concept of spikes in the proposed method. Also, is the considered algorithm generic for all modulations?

6. The lack of comparison with other methods, analytical results, as well as complexity analysis makes the paper has limited contributions.

Author Response

Please see the attached response. 

Round 2

Reviewer 2 Report

The authors have addressed my concerns, no further comment.

Author Response

Thank you very much for your positive feedback. 

Reviewer 3 Report

All problems have been settled.

Author Response

(The authors gave the same response as above.)

Reviewer 4 Report

I would like to thank the efforts of the authors for the revision, however, the manuscript is still on the boundary of acceptance.

1. General comment one: please consider improving the quality of the response letter. It is not easy to go through the draft and find your responses. Also, some revisions might need more descriptions/motivations, which is the purpose of the response letter.

2. General comment two: the figures in the manuscript should show at the place where they are first mentioned in the text.

3. About my previous comment 2: "The source, relay, and destination all have a single antenna, which significantly limits the contribution of the work." I could not see how it is easy to be extended to multiple-antenna systems.

4. The direct link is highlighted in (2) but I could not find good motivations.

5. About my previous comment 6: "The lack of comparison with other methods, analytical results, as well as complexity analysis makes the paper has limited contributions." The improvement in the revision, e.g., comparison with [10], is not sufficient to cover this concern.
